# Long Non-Coding RNAs and Proliferative Retinal Diseases

**DOI:** 10.3390/pharmaceutics15051454

**Published:** 2023-05-10

**Authors:** Anamika Sharma, Nikhlesh K. Singh

**Affiliations:** 1Integrative Biosciences Center, Wayne State University, Detroit, MI 48202, USA; anamika.sharma@wayne.edu; 2Department of Ophthalmology, Visual and Anatomical Sciences, Wayne State University School of Medicine, Detroit, MI 48202, USA

**Keywords:** LncRNAs, diabetic retinopathy, AMD, retinopathy of prematurity, retinal vein occlusion, proliferative vitreoretinopahty

## Abstract

Retinopathy refers to disorders that affect the retina of the eye, which are frequently caused by damage to the retina’s vascular system. This causes leakage, proliferation, or overgrowth of blood vessels through the retina, which can lead to retinal detachment or breakdown, resulting in vision loss and, in rare cases, blindness. In recent years, high-throughput sequencing has significantly hastened the discovery of new long non-coding RNAs (lncRNAs) and their biological functions. LncRNAs are rapidly becoming recognized as critical regulators of several key biological processes. Current breakthroughs in bioinformatics have resulted in the identification of several lncRNAs that may have a role in retinal disorders. Nevertheless, mechanistic investigations have yet to reveal the relevance of these lncRNAs in retinal disorders. Using lncRNA transcripts for diagnostic and/or therapeutic purposes may aid in the development of appropriate treatment regimens and long-term benefits for patients, as traditional medicines and antibody therapy only provide temporary benefits that must be repeated. In contrast, gene-based therapies can provide tailored, long-term treatment solutions. Here, we will discuss how different lncRNAs affect different retinopathies, including age-related macular degeneration (AMD), diabetic retinopathy (DR), central retinal vein occlusion (CRVO), proliferative vitreoretinopathy (PVR), and retinopathy of prematurity (ROP), which can cause visual impairment and blindness, and how these retinopathies can be identified and treated using lncRNAs.

## 1. Introduction

Ocular disorders are prevalent in both developed and developing nations. In the world, at least 2.2 billion people suffer from some form of vision impairment, with at least 1 billion cases being either preventable or unaddressed [1]. From 2020 to 2045, the global DR population is expected to grow by 55.6% (57.4 million) [2]. The term “retinopathy” refers to a broad range of disorders that can result in retina-related vision loss. A normal, healthy retina contains blood vessels that transport oxygen and nutrients, so blood supply to the retina is vital. Retinopathy is a condition in which blood vessels leak, overrun, or grow through the retina. Detachment or breakdown of the retina can cause vision loss and, in rare circumstances, blindness. Visual impairment and blindness can be caused by diabetic retinopathy (DR), age-related macular degeneration, retinopathy of prematurity (ROP), and proliferative vitreoretinopathy. All these retinopathies negatively impact human health and general wellbeing and impose a significant psychological, medical, and economic strain on patients and community. Preventing visual impairment through early detection and treatment is critical worldwide. Many human diseases have been linked to long non-coding RNAs (lncRNAs), making them potential effectors and treatment options [3]. Long non-coding RNAs (lncRNAs) have recently been discovered to be differentially expressed in the retina, and play an essential role in various retinopathies, such as diabetic retinopathy, age-related macular degeneration, proliferative vitreoretinopathy, retinal vein occlusion, and others.

## 2. LncRNA and Their Regulation

Historically, it was thought that a significant portion of the human genome that did not code for proteins was junk DNA. Over the last decade, the advancement in next-generation sequence analysis has enabled researchers to conduct in-depth investigations of the genome’s non-coding regions at an unprecedented resolution and scale [3]. Although protein-coding genes account for a small portion of the human genome, nearly the entire genome is transcribed [4]. Long non-coding RNAs (lncRNAs) are made up of more than 200 nucleotides. Despite having 5′ capped ends and being spliced similarly to mRNAs, lncRNAs lack open reading frames (ORFs), implying that they lack protein-coding potential. LncRNA expression is not as consistent as mRNA expression and can vary, depending on the species, tissue, or cell [5]. Many lncRNAs, in contrast to mRNAs, have modest levels of expression, poor evolutionary conservation, and tissue- or cell-specific expression [6]. There are less transcription factor (TF) binding motifs in lncRNA promoters, according to studies, and TF binding events are inconsistent with this finding [7]. LncRNAs are involved in many different cellular processes, such as the control of transcription, post-transcriptional regulation (such as splicing), organellar and structural organization of the cell, and genome integrity [6]. Depending on where they are found, lncRNAs may have different roles. Around 15% of lncRNAs are in the cytoplasm, while the nucleus contains the majority of them [8]. LncRNAs located in the cell’s nucleus can affect how genes are expressed, whereas lncRNAs found in the cytoplasm can behave as miRNA sponges [9]. LncRNAs fall within the category of regulatory non-coding RNAs, due to their structural features. LncRNAs are made up of molecules that serve as scaffolds, decoys, guides, and signals [10]. In animals, RNA polymerase II produces the great majority of lncRNAs; however, some lncRNAs are also produced by RNA polymerase I or III [11,12]. Long non-coding RNAs are classified into intronic lncRNAs, intergenic lncRNAs, bidirectional lncRNAs, antisense lncRNAs, and sense lncRNAs (Figure 1) [10]. LncRNA genes are distributed throughout the genome, but tend to congregate in introns, exons, and regulatory regions (such as enhancers and promoters) of other genes. Figure 2 depicts the functional classification of lncRNAs. They can take on both open (linear) and closed (circular) structures. Several studies have revealed that lncRNAs are tissue-specific in the eye and are crucial in the pathophysiology of several eye diseases [13,14].

## 3. LncRNAs and Proliferative Retinopathies

LncRNAs are becoming essential regulators of several critical biological pathways. Although it is known that a variety of lncRNAs are explicitly expressed in the developing retina, it is unknown what function these lncRNAs serve in this organ. Several lncRNAs have been discovered recently, due to advances in bioinformatics. These lncRNAs may play a significant role in retinal diseases; however, mechanistic studies still need to be conducted to determine their functional importance.

### 3.1. LncRNAs and AMD

Age-related macular degeneration, a neurodegenerative eye disease, is the most prevalent cause of legal blindness and visual loss in older individuals. Over 300 million people are expected to be affected globally by it in 2040 [16]. The macula, a tiny functional region in the center of the retina that controls both fine and color vision, is harmed by AMD. The disease impairs vision by gradually damaging and destroying photoreceptors and the underlying retinal pigment epithelium (RPE) [17]. Ventral anterior homeobox 2 opposite strand isoform 1 or 2(Vax2os1/2) lncRNAs contribute to wet age-related macular degeneration by altering the balance of various angiogenic factors in the eye [18]. Increased expression of Vax2os1 and Vax2os2 lncRNAs was observed in CNV patients. The altered expression of Vax2os1 causes cell cycle changes in photoreceptor progenitor cells [17]. Specifically, due to Vax2os1 overexpression during the early postnatal mouse retinal development, photoreceptor progenitors had poor cell cycle progression and retarded differentiation processes. Vax2os1 overexpression slows cell cycle development and differentiation in mouse photoreceptor-derived 661W cells. Vax2os1 lncRNA controls cell cycle progression during mammalian retina development [19].

An amount of 217 lncRNAs, with varied levels of expression, were observed in the microarray analysis of human RPE cells derived from an induced pluripotent stem cell (iPSC) [20]. The zinc finger protein 503 antisense RNA 1 (ZNF503-AS1) lncRNA was downregulated in the RPE-choroid of people with atrophic AMD and upregulated in the RPE-cytoplasm during RPE differentiation [20]. ZNF503-AS1 deficiency was found to suppress differentiation of iPSC-derived RPE cells in vitro, while simultaneously encouraging proliferation and migration. ZNF503 is a transcriptional repressor that can bind both DNA and RNA [20]. In addition, nuclear factor kappa B (NFκB) has been identified as a possible transcription factor for ZNF503-AS1. RPE cell dedifferentiation play a significant role in the pathogenesis of dry AMD, making the lncRNA ZXNF503-AS1 an attractive biomarker and therapeutic target [20].

The lncRNA LINC00167 expression was decreased in AMD patients’ defective RPE cells and RPE-choroid samples [21]. Furthermore, it was also discovered that elevated RPE cell proliferation was associated with upregulated LINC00167 expression. Studies conducted in vitro demonstrated that decreasing LINC00167 expression led to the dedifferentiation of RPE cells, an increase in the formation of mitochondrial ROS, and a reduction in phagocytic activity. Moreover, LINC00167 served as a sponge for miR-203a-3p to enhance the transcription of the SOCS3 (suppressor of cytokine signaling 3), which inhibits the JAK/STAT signaling pathway. Overall, it was reported that LINC00167 decreases AMD progression by maintaining RPE cell differentiation via modulation of the LINC00167/miR-203a-3p/SOCS3 axis [21]. It was also suggested that lncRNAs ZNF503-AS1 and LINC00167 can prevent and slow the progression of AMD by inhibiting RPE cell dedifferentiation [21].

The significance of lncRNAs in the emergence of AMD was explored by examining the expression profile of early AMD patients [22]. The expression profile analysis of this study revealed that, out of 266 differentially expressed genes, 64 genes code for long non-coding RNAs. The above research also suggested that these differentially expressed lncRNAs are essential in visual perception, the sensory experience of light stimulation, and cognitive processes. Furthermore, Zhu et al. discovered that the level of RP11-234O6.2 lncRNA was decreased in the aging RPE cells (ARPE-19 cell line) [22]. The results of a forced overexpression of RP11-234O6.2 showed that aging RPE cells had increased cell viability and decreased apoptosis because of the experiment. The findings also suggest that lncRNAs have distinct expression patterns in early AMD, and that lncRNAs, such as RP11-234O6.2, regulate early AMD, and may have therapeutic applications.

Neurovascular dysfunction alters lncRNA MIAT expression. MIAT regulates vascular permeability in the retina and cornea. MIAT via miR-150-5p/VEGF network regulates neural and vascular cell function [23]. The inhibition of lncRNA MALAT1 resulted in a decrease in human retinal microvascular endothelial cell proliferation, migration, and angiogenesis. This was accomplished by targeting miR-125b for VE-cadherin/-catenin complex inhibition. These findings have significant implications for retinal neoangiogenesis, and point to MALAT1 as a possible therapeutic target in disorders associated to retinal neoangiogenesis, such as age-related macular degeneration (AMD) [24]. The lncRNA HDAC4-AS1 functions as a link between the transcriptional activities of HIF-1α and HDAC4. In ARPE-19 cells subjected to hypoxia, lncRNA HDAC4-AS1 was able to decrease the production of HDAC4 through the regulation of HIF-1α. Since hypoxia-inducible transcription factors were found to be related with CNV and the progression of AMD, it is possible that this regulation plays a significant role in the development of AMD [25]. Figure 3 depicts the lncRNAs implicated in AMD pathogenesis.

### 3.2. LncRNAs and Diabetic Retinopathy

Diabetes retinopathy (DR) is a primary cause of adult blindness globally. It is the most common diabetic microvascular problem. The impact of lncRNAs on DR has been extensively studied. No DR, mild non-proliferative diabetic retinopathy (NP-DR), moderate NP-DR, severe NP-DR, and proliferative DR are the five phases of DR, according to the International Council of Ophthalmology. According to a 2020 meta-analysis, 22.27% of patients with diabetes have DR, 6.1% have sight-threatening DR, and 4.07% have substantial clinical macular edema [2]. The number of adults with DR is anticipated to climb by 25.9% (129.84 million) in 2030 and 55.6% (160.50 million) in 2045 [2].

The lncRNA-MALAT1 was shown to be highly increased in animal models of diabetic retinopathy. Knocking down lncRNA-MALAT1 was proven to help cure streptozotocin-induced diabetic retinopathy in rats. Moreover, the VEGF-regulated proliferation, migration, and tube formation of retinal endothelial cells were reduced by MALAT1 knockdown [26]. LncRNA-MALAT1 regulate endothelial cells function by controlling the expression of S-phase cyclins (cyclinA2, cyclin B1, and cyclin B2), and cell cycle inhibitory genes (p21 and p27Kip1) [27]. According to Zhang et al. [28], retinal progenitor cells have significant levels of lncRNA-MIAT expression, which serves as a sponge for miR-150. LncRNA-MIAT affects endothelial cells and diabetic retinopathy by increasing the production of VEGF, a miR-150-5p target gene. Also, by blocking tumor necrosis factor and intercellular adhesion molecule 1, lncRNA-MIAT knockdown can reduce vascular leakage and inflammation [29]. In high glucose conditions, lncRNA MALAT1 via miR-378a-3p upregulates PDE6G expression, promoting retinal vascular endothelial cell proliferation and inhibiting apoptosis [30]. In contrast, by modulating the miR-195/mfn2 axis, a prospective target for DR treatment, lncRNA SNHG16 inhibits oxidative stress-induced pathological angiogenesis in EC [31]. Proliferative DR had higher SNHG16 and E2F1 expression and lower levels of miR-20a-5p than non-proliferative or control DR. The lncRNA SNHG16 controls E2F1 expression through miR-20a-5p, exacerbating PDR [32]. LncRNA H19, through miR-200b, inhibits TGF-1 signaling protein expression to prevent endothelial-mesenchymal transition in DR [33]. LncRNA MIR497HG targets the miRNA-128-3p/SIRT1 axis to prevent HREC growth and migration [34]. In response to high glucose stress, RNCR3 is substantially upregulated, both in vitro and in vivo. RNCR3 depletion restores retinal vascular integrity and decreases vascular leakage, and inflammation. lncRNA RNCR3 depletion reduces retinal endothelial cell migration and proliferation through the RNCR3/KLF2/miR-185-5p pathway, improving retinal vascular integrity [35].

Recently, researchers discovered that lncRNA ANRIL regulates the expression and activity of VEGF in diabetic retinopathy [36]. The brain-derived neurotrophic factor (BDNF) protects retinal ganglion cells (RGCs) in diabetic retinopathy, and its antisense RNA (BDNF-AS) is a lncRNA that decreases BDNF mRNA expression [37]. Directly binding to BDNF, BDNF-AS can control BDNF-regulated cytokine expressions, such as IL-2 and IL-6. Therefore, inhibiting BDNF-AS can protect RGCs from ischemia-induced damage. This might be an alternative method of treating DR [37].

The maternally expressed gene 3 (MEG3) is a tumor repressor lncRNA [38]. The expression of lncRNA-MEG3 was decreased in endothelial cells subjected to hydrogen peroxide, and in streptozotocin-induced diabetic mice [39]. Depleting lncRNA-MEG3 increases the phosphorylation of PI3K and Akt, suggesting its role in angiogenesis. Further study into how lncRNA-MEG3 and PI3K/Akt signaling interact may aid in understanding how blood vessels develop in the retina [40]. Another prominent symptom of diabetes-related retinal neurodegeneration is RGC damage. LncRNA-SOX2OT levels in RGCs subjected to high glucose or oxidative stress decreased with time. Moreover, nuclear factor erythroid 2-related factor 2 (NRF2) and heme oxygenase-1 (HO-1) can shield cells from oxidative stress. The possibility that lncRNA-SOX2OT might be employed to treat DR arises from the finding that lncRNA-SOX2OT can activate the NRF2/HO-1 signaling pathway [41]. The expression of lncRNAs, such as H19, ANRIL, HULC, HOTAIR, WISPER, MIAT, and ZFAS1, increased significantly in serum samples from diabetic patients at various phases of DR, but not MALAT1 or MEG3. The expression patterns of ANRIL and three others circulating lncRNAs (RNCR2, MALAT1, and PVT1) effectively distinguished between diabetes mellitus, controls, and DR, even though none of these lncRNAs were associated with the development of DR or the response to anti-VEGF medication [42].

The serum levels of lncRNA HOTAIR and MALAT1 were significantly higher in non-proliferative DR and PDR patients than in healthy controls. Moreover, Receiver-Operating Characteristic (ROC) analysis revealed that these lncRNAs could distinguish between NPDR, PDR, and non-DR controls [43]. Clinical samples from individuals with diabetic retinopathy, Müller cells treated with high glucose, and diabetic retinas from a rat model all included much greater levels of lncRNA AQP4-AS1. In vivo reduction of AQP4-AS1 helped minimize vascular dysfunction and retinal neurodegeneration, as observed by decrease in RGC loss and reactive gliosis. Endothelial cell and RGC activities were modified indirectly by AQP4-AS1, whereas Müller cell functions were modified directly. AQP4-AS1 controls retinal neurovascular dysfunction by regulating AQP4 expression [44]. The lncRNA ZNF503 regulates retinal pigment epithelial cell differentiation, and increased plasma lncRNA ZNF503-AS1 levels were linked to diabetic retinopathy. It was reported that diabetic retinopathy patients had higher plasma lncRNA ZNF503-AS1, overexpression of ZNF503-AS1 inhibited proliferation, induced apoptosis, and increased TGF-β1 signaling [45]. Compared to plasma samples from healthy controls, LINC00673 was downregulated in DR patients. In retinal pigmental epithelial cells (RPECs), hyperglycemia downregulated LINC00673 and overexpression of LINC00673 reduced p53 and RPEC apoptosis [46].

In diabetic retinopathy, MIAT lncRNA controls microRNA-29 to modify cell death [35]. In animal models of diabetic retinopathy, MIAT also created a feedback loop between VEGF and miR-150-5p to control endothelial cell activity [47]. The XIST levels were lower in the retinal tissues of the streptozotocin-induced DR mice and the high glucose (HG)-induced human muller cells. XIST overexpression suppressed pro-inflammatory cytokines and HG-induced mouse retinal muller cells and human retinal muller cells activation. By interacting with SIRT1 and blocking SIRT1 ubiquitination, XIST increased SIRT1 expression. Furthermore, XIST overexpression impacted the activation of murine and human retinal muller cells, as well as the high glucose-induced expression of pro-inflammatory cytokines. At the same time, SIRT1 inhibition partially counteracted this effect [48]. By directly binding to and inhibiting the expression of hsa-miR-21-5p in retinal pigmental epithelial cells, lncRNA-XIST prevents its apoptosis and restores the ability to migrate in response to high glucose [49]. The expression of serum LncRNA-OGRU was considerably elevated in DR patients compared to healthy individuals. OGRU knockdown dramatically decreased the expression of VEGF and TGF-1 in high glucose-incubated Müller cells. OGRU knockdown in vitro reduced high glucose-induced inflammatory response and oxidative stress by decreasing NF-κB signaling pathways and increasing nuclear factor erythroid 2-related factor 2 (Nrf2) [50]. In high glucose-stimulated Müller cells, miR-320 overexpression reduced TGF-β1 signaling, inflammation, and ROS generation. High glucose-treated Müller cells also showed a significant decrease in ubiquitin-specific peptidase 14 (USP14) expression after OGRU knockdown or miR-320 overexpression. As a result, the lncRNA-OGRU/miR-320/USP14 axis may thus serve as a therapeutic target for the treatment of DR. The lncRNA-OGRU/miR-320/USP14 axis may be a therapeutic target for the treatment of DR [50]. Under high glucose circumstances, downregulation of lncRNA NEAT1 raised miR-497 concentration, which lowered brain-derived neurotrophic factor production, hence promoting Müller cell death [51]. LncRNA SNHG7 adsorbs miR-34a-5p and adversely regulates it. The overexpression of lncRNA SNHG7 regulates HRMVECs tube formation and high glucose-induced endothelial-mesenchymal transition (EndMT). To treat diabetic retinopathy, exosomal lncRNA SNHG7 supplementation from mesenchymal stem cells, boosting its concentration in vivo, or targeting the miR-34a-5p/XBP1 axis might be employed [52]. The lncRNAs involved in DR progression are shown in Figure 4.

### 3.3. LncRNAs and Proliferative Vitreoretinopathy

Proliferative vitreoretinopathy (PVR) is a potentially fatal consequence of a rhegmatogenous retinal detachment. The RPE, glia, fibroblasts, and inflammatory cells influence the growth of PVR. PVR patients had the greatest RPE levels in their pre-retinal membranes [53]. A microarray investigation revealed 78 lncRNAs that were abnormally expressed in PVR patients’ epiretinal membranes (ERMs). Forty-eight lncRNA transcripts were up-regulated, and 30 were down-regulated among them [54]. Silencing MALAT1 suppresses TGF1-induced RPE cell epithelial-mesenchymal transition, migration, and proliferation by increasing Smad2/3 signaling [55]. LncRNA-MALAT1 was considerably increased in the cellular and plasma fractions of PVR patients’ peripheral blood, but decreased following PVR surgery. In the future, MALAT1 siRNA (siMALAT1) might be injected intravenously, with the aim of gene therapy, to lower MALAT1 expression [54]. These results collectively imply that lncRNAs can control PVR pathogenesis. The LINC01705-201 (lncRNA in RPE) is involved in TGF-1-induced epithelial-to-mesenchymal transition of human RPE cells and PVR [56].

### 3.4. LncRNAs and Retinopathy of Prematurity

Premature babies are susceptible to the progressive retinal vascular disease known as retinal retinopathy of prematurity (ROP). The aberrant development of retinal vessels characterizes ROP. The prevalence of ROP has steadily grown over time and is currently the leading cause of blindness in children [57,58]. Many lncRNAs have been studied for their impact in ROP. Two studies showed that the lncRNA MALAT1 plays a pro-angiogenic role [59,60]. The expression of MALAT1 lncRNA went up in murine model of oxygen-induced retinopathy. MALAT1 inhibition decreased retinal neovascularization and shut down the Akt/VEGF pathway and expression of inflammatory cytokines, such as IL-1, IL-6, and TNF-α. These findings imply that lncRNA MALAT1 inhibition may reduce the development of ROP [58]. LncRNA MALAT1 can modulate early growth response by acting as a sponge for miR-124-3p (EGR1) [60]. The lncRNA MALAT1 and EGR1 were predicted to be the interaction partners of miR-124-3p, using bioinformatics analysis. The EGR1 and lncRNA MALAT1 expression was increased in hypoxic HUVECs and OIR-induced mouse retinas. MALAT1 inhibition or miR-124-3p overexpression reduced hypoxia-induced EGR1 levels in HUVECs. In vitro and in vivo models revealed a novel regulatory axis for the lncRNA MALAT1/miR-124-3p/EGR1. LncRNA MEG3, a maternally expressed gene, is involved with ROP. In OIR mice models, intravitreal injection lentivirus overexpressing MEG3 reduced retinal angiogenesis via the VEGF/PI3K/Akt signaling pathway, and suppressed inflammatory indicators, such as IL-1 and IL-6 [61]. The intravitreal injection of lncRNA MIAT inhibited retinal angiogenesis by decreasing the VEGF/PI3K/Akt pathway [62]. As a result, lncRNA MEG3 or MIAT may be a suitable therapeutic target for ROP. TUG1 LncRNA controls the production of numerous factors in endothelial cells by competitive adsorption of a range of microRNAs (miRNAs), and, hence, plays a role in angiogenesis and vascular remodeling of endothelial cells [63]. TUG1 expression changes are crucial in neurovascular disease [64]. The TUG1 expression levels were induced in the hypoxic retinas of mice treated with OIR. LncRNA TUG1 depletion reduces OIR-induced retinal neovascularization, apoptosis, inflammation, and VEGFA expression in mice. Furthermore, it was shown that the lncRNA TUG1 regulates VEGFA expression and acts as a sponge for miR-299-3p [65].

## 4. Future Perspectives

LncRNAs are a hot topic in clinical medicine, indicating a growing fascination towards epigenetic regulation of expression as a critical mechanism regulating healthy and disease related phenotypes. Several lncRNAs have been found that may play critical roles in human physiology and disease states. In the long term, harnessing these transcripts for diagnostic and therapeutic reasons may help patients obtain the best treatment possible, given the relevance and involvement of lncRNAs in disease progression. The effects of lncRNAs on various proliferative retinopathies is summarized in Table 1.

Although the relevance of lncRNAs in human diseases is in its early stages, some lncRNAs have been reported as disease prognostic markers. Several of these lncRNAs have disease-associated single nucleotide polymorphisms, implying that they could be used in molecular diagnostics. For example, LncRNA-SNHG16 expression is higher in diabetic retinopathy patients’ peripheral blood samples than in controls [68], though the test specificity of lncRNAs in human diseases is still debatable.

Targeting lncRNAs seems plausible, since more mRNA-targeting antisense oligonucleotide medications are entering the clinics, due to research on the disease-relevant functions of human lncRNAs [69]. Typically, these RNA inhibitors are made of synthetic nucleic acid mimics that base-pair with RNA, such as peptide nucleic acid, locked nucleic acid, or phosphorodiamidate morpholino. Cell-penetrating peptides or lipid nanoparticles can be used to accomplish cellular delivery [32]. Melatonin, by enhancing the lncRNA-MEG3 expression, inhibits proinflammatory cytokine production by Müller cells in experimental diabetic retinopathy models [70]. Hawthorn polyphenol extracts decrease high glucose-induced RPE cell apoptosis and inflammatory responses [71]. Niaspan via miR-126 regulation restore damaged blood vessels in the retina and improve diabetic retinopathy [72]. By increasing miR-26a expression, ginsenoside Rg1 prevents the high glucose-induced apoptosis of RPE cells [73]. Replication experiments, with larger cohorts and advanced knowledge of the molecular roles of lncRNA markers, are required before the potential of lncRNAs in personalized treatment can be fully realized.

Gene-based therapies can provide tailored treatments for long-term therapeutic cures, in contrast to traditional medicines and antibody therapies, which only have short-term advantages and must be repeated. Even though research on lncRNAs in ocular illnesses has not received as much attention as research on cancer, neuropathy, and other diseases, the evidence that is now available demonstrate their important roles and the promise for gene therapy in these conditions. LncRNAs have a lot of potential as biomarkers for molecular diagnosis and treatment.

LncRNAs have emerged as a focus of current research because the majority of their functions are still unknown. However, several issues must be addressed before they can be used in clinical practice. Targeting a single lncRNA may have unforeseen effects, because each lncRNA has many targets. Many lncRNAs’ functions and regulatory mechanisms in ocular tissues require further investigation. Most gene therapy research for ocular diseases is done in vitro on cell lines, or in vivo in animal experiments. Still, only a few studies have reported that lncRNA knockout animal show no change in phenotype. LncRNA therapy’s clinical use will develop if the concerns regarding safety and efficacy are further elucidated. Therefore, further research is needed to fully understand the networks regulated by lncRNAs linked to ocular illnesses. Before going for clinical trials, a more thorough analysis of preclinical studies is required.

## Figures and Tables

**Figure 1 pharmaceutics-15-01454-f001:**
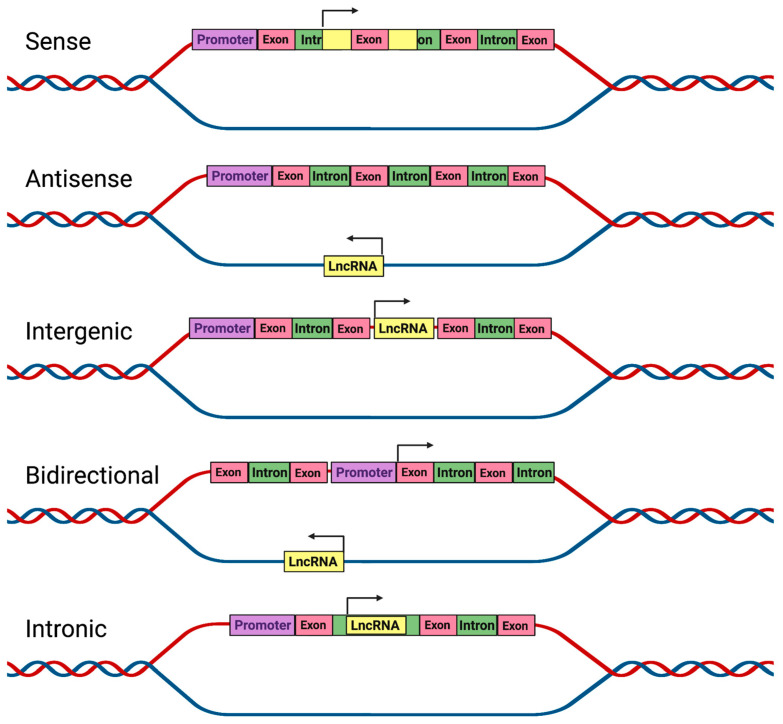
Classification of lncRNAs. LncRNAs are categorized based on their gene location, in relation to protein-coding genes. LncRNAs are classified into sense lncRNAs (contain exons from protein-coding genes), antisense lncRNAs (transcribed from the opposite strand of protein-coding or non-coding genes), intergenic lncRNAs (do not overlap with any protein-coding gene), bidirectional lncRNAs (transcribed from the opposite direction of a protein-coding gene with same promoter), and intronic lncRNAs (transcribed from protein-coding genes of the introns).

**Figure 2 pharmaceutics-15-01454-f002:**
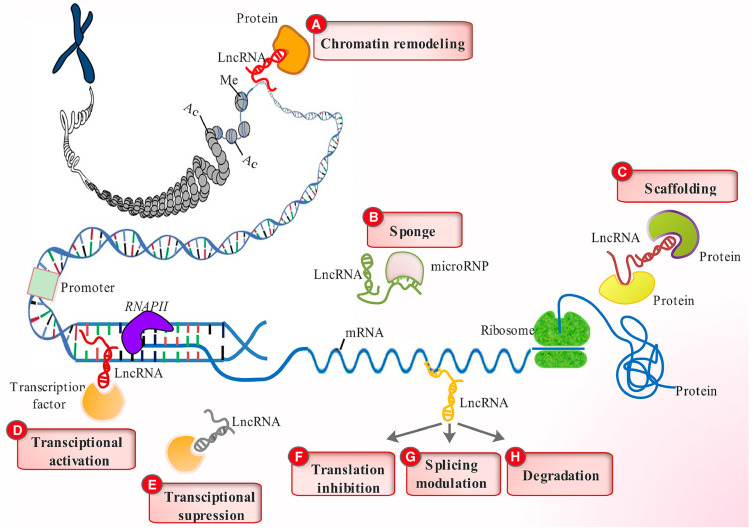
LncRNA regulatory functions. LncRNAs engage various chromatin-remodeling complex proteins to alter chromatin organization (**A**), function as sponges to limit miRNA effects (**B**), act as scaffolds for protein docking (**C**), stimulate gene transcription by directing transcription factors to their promoters (**D**), decrease transcription by sequestering and preventing transcription factors from binding to their promoters (**E**), influence mRNA function by limiting translation (**F**), modify splicing patterns (**G**), and expose them to degradation (**H**). Reprinted with permission from [15]. Copyright John Wiley and Sons, 2017.

**Figure 3 pharmaceutics-15-01454-f003:**
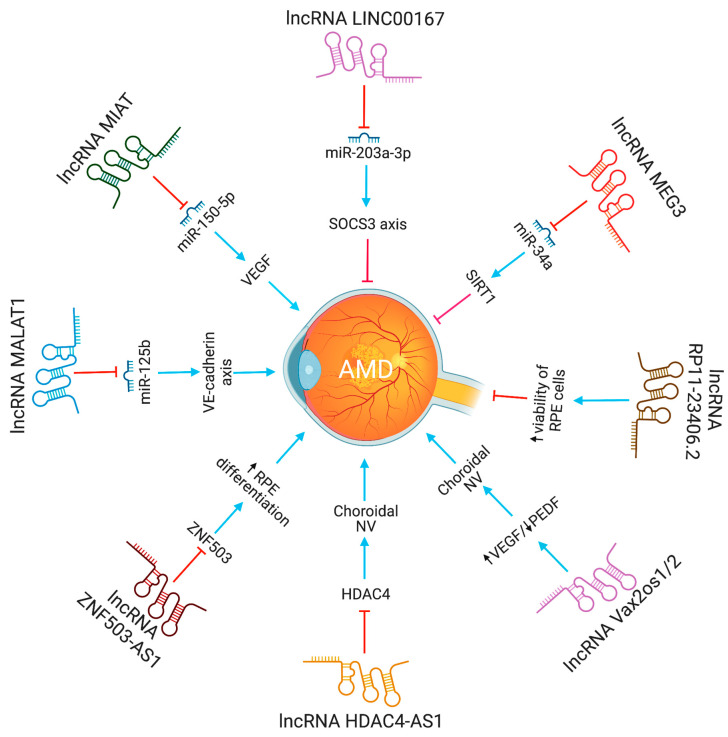
The functions and biological pathways of lncRNAs in age-related macular degeneration (AMD). Blue arrows represent upregulation or activation, while red inhibitory signs represent downregulation or inhibition.

**Figure 4 pharmaceutics-15-01454-f004:**
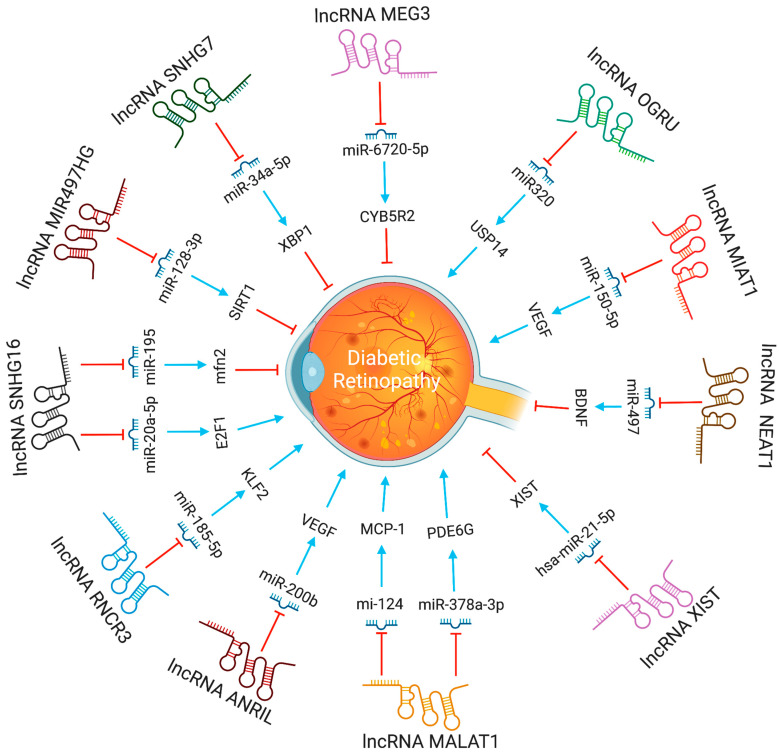
The functions and biological pathways of lncRNAs in diabetic retinopathy. Blue arrows represent upregulation or activation, while red inhibitory signs represent downregulation or inhibition.

**Table 1 pharmaceutics-15-01454-t001:** Effects of long non-coding RNAs (lncRNAs) in proliferative retinal diseases.

Disease	LncRNA	Role	Upregulated or Downregulated	References
Age-Related Macular Degeneration	Vax2os1/Vax2os2	Induces expression of proangiogenic factors in the eye, leading to wet AMD	Upregulated	[18,19]
	ZNF503-AS1	Enhances retinal pigment epithelial (RPE) cell dedifferentiation and AMD	Upregulated	[20]
	MIAT	Regulates vascular and neural cell function via modulating miR-150-5p/VEGF pathway, leading to AMD	Upregulated	[23]
	MALAT1	Inhibits VE-cadherin/catenin complex via miR-125b, leading to endothelial dysfunction and AMD	Upregulated	[24]
	HDAC4-AS1	Increases HIF-1a mediated CNV, leading to AMD	Upregulated	[25]
	RP11-145 234O6.2	Enhances RPE cell viability, leading to inhibition of AMD	Downregulated	[22]
	LINC00167	Inhibits RPE dedifferentiation and AMD progression	Downregulated	[21]
	MEG3	Enhances SIRTP1 expression and AMD progression	Downregulated	[66]
Diabetic Retinopathy	MALAT1	Induces PDE6G expression via miR-378a-3p and regulates DR	Upregulated	[30]
	MIAT	Induces VEGF expression and DR	Upregulated	[29]
	RNCR3	Enhances DR by increasing retinal EC migration and proliferation	Upregulated	[35]
	ANRIL	Regulates VEGF expression and enhances DR	Upregulated	[36]
	BNDF-AS	Decreases BNDF, IL-2, and IL-6 expression, resulting in DR enhancement	Upregulated	[37]
	AQP4-AS1	Enhances DR by regulating AQP4 expression, which controls retinal neurovascular dysfunction	Upregulated	[44]
	ZNF503-AS1	Induces TGF-β1 signaling and DR	Upregulated	[45]
	OGRU	Enhances DR by regulating the miR-320/USP14 axis	Upregulated	[50]
	SNHG16	Enhances PDR by regulating E2F1 expression	Downregulated	[32]
	H19	Prevents EMT and inhibits DR	Downregulated	[33]
	MIR497HG	Reduces HREC growth and migration and inhibit DR	Downregulated	[34]
	MEG3	Decreases angiogenesis and DR	Downregulated	[40,67]
	SOX2OT	Activates NRF2/HO-1 signaling and inhibits DR	Downregulated	[41]
	XIST	Increases RPE cell viability and reduces DR progression	Downregulated	[48,49]
	LINC00673	Inhibits DR by reducing p53 expression and RPE apoptosis	Downregulated	[46]
	NEAT1	Enhances BNDF expression and reduces DR progression	Downregulated	[51]
	SNHG7	Inhibits DR by regulating miR-34a-5p/XBP1 axis	Downregulated	[52]
Retinopathy of Prematurity	MALAT1	Enhances expression of proangiogenic factors and ROP	Upregulated	[59,60]
	TUG1	Induces VEGFA expression and ROP	Upregulated	[65]
	MIAT	Inhibits ROP by downregulating VEGF/PI3K/Akt signaling	Downregulated	[62]
	MEG3	Inhibits ROP by downregulating VEGF/PI3K/Akt signaling	Downregulated	[61]

## Data Availability

Data available in a publicly accessible repository.

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
