# Peer review of "Long Non-Coding RNAs and Proliferative Retinal Diseases"

_pharmaceutics, 2023, doi:10.3390/pharmaceutics15051454_

Round 1

Reviewer 1 Report

In the review by Sharma and Singh entitled: " Long Non-coding RNAs and Proliferative Retinal Diseases", the authors discuss about new long noncoding RNAs (lncRNAs) and their biological functions in different ocular diseases, including AMD and diabetic retinopathy, for diagnostic and therapeutic purposes. Given the relevance and involvement of lncRNAs in ocular disease progression, they claim that LncRNAs have a lot of potential as biomarkers for molecular diagnosis and treatment. However, the majority of their functions of lncRNAs in ocular diseases are still unclear, it is still too soon to claim lncRNAs  as potential effectors and treatment options for different kinds of ocular diseases.    

I think the review discuss an important and interesting topic and should be considered for publication after revision suggested below:  

Major Revision

Given little literatures are currently available on roles of lncRNAs in proliferative vitreoretinopathy and central retinal vein occlusion, sections 3.3 and 3.4 on pages 9 and 10 should be removed from the review.      

Author Response

I think the review discuss an important and interesting topic and should be considered for publication after revision suggested below:  

Major Revision

Given little literatures are currently available on roles of lncRNAs in proliferative vitreoretinopathy and central retinal vein occlusion, sections 3.3 and 3.4 on pages 9 and 10 should be removed from the review.      

Answer: In response to Reviewer #1’s suggestions, we have removed section 3.4 of the manuscript. We have retained section 3.3 in the revised manuscript, as we have discussed the role of LINC01705-201 and LncRNA-MALAT1 on proliferative vitreoretinopathy in this section.

Reviewer 2 Report

In this study, the authors described how different long non-coding RNAs (lncRNAs) affect different retinopathies. The authors divided the manuscript into 4 sections, in which the relationship between lncRNAs and biology of each proliferative retinopathy is discussed. Overall, current understandings related to lncRNAs and retinopathies are well summarized. Therefore, the manuscript should be published after minor revisions.

1. The authors should summarize upregulated lncRNAs and/or downregulated lncRNAs in the retinal diseases the authors focused on in a Table. It is important to show which lncRNAs augment and/or abrogate development of the diseases. 

2. Are there any disease-specific lncRNAs among the diseases? 

3. The authors had better describe the clinical benefit of utilizing the information about lncRNA expression in such diseases in terms of diagnosis and treatment. 

Author Response

1. The authors should summarize upregulated lncRNAs and/or downregulated lncRNAs in the retinal diseases the authors focused on in a Table. It is important to show which lncRNAs augment and/or abrogate development of the diseases. 

Answer: In response to Reviewer #2’s suggestions, we have now included a table in the revised manuscript (please refer to Table 1).

2. Are there any disease-specific lncRNAs among the diseases? 

Answer: Although the relevance of lncRNAs in human diseases is still in its early stages, some lncRNAs have been reported as disease prognostic and risk stratification markers. Several of these lncRNAs have disease-associated single nucleotide polymorphisms, implying that they could be used in molecular diagnostics. For example, LncRNA-SNHG16 expression is higher in diabetic retinopathy patients' peripheral blood samples than in controls (Can. J. Physiol. Pharmacol. 2021, 99, 1207-1216), though the test specificity of lncRNAs in human diseases is still debatable. It is now included in the revised manuscript (please refer to Page 10, lines 336-341).

3. The authors had better describe the clinical benefit of utilizing the information about lncRNA expression in such diseases in terms of diagnosis and treatment. 

Answer: In response to Reviewer #2’s suggestions, we have now included it in the revised manuscript (please refer to Page 12, lines 410-423).

Reviewer 3 Report

This is a nice review with long noncoding RNAs (lncRNAs) in specific topic of Retinopathy. It was a pleasure to read and very informative.

My only comment is that there are so many abbreviations that it takes time to go back in the manuscript to find the definition. If the authors think it would be useful for the readers to have  a table with abbreviations and definitions.  I leave this up to the author’s discretion.

The graphics are very nicely done. Very helpful.

Author Response

My only comment is that there are so many abbreviations that it takes time to go back in the manuscript to find the definition. If the authors think it would be useful for the readers to have a table with abbreviations and definitions.  I leave this up to the author’s discretion.

The graphics are very nicely done. Very helpful.

Answer: We are thankful to Reviewer #3 for his/her encouraging comments. An abbreviations table is now included in the revised manuscript.

Round 2

Reviewer 1 Report

The authors have addressed my concerns.